

# Sea Ice Concentration Estimates from ICESat-2 Linear Ice Fraction. Part 2: Gridded Data Comparison and Bias Estimation.

Christopher Horvat[1], Ellen Buckley[2], and Madelyn Stewart[3]

[1]Department of Earth, Environmental, and Planetary Sciences, Brown University, Providence, USA
[2]Department of Earth Science and Environmental Change, University of Illinois Urbana-Champaign, Champaign, IL, USA
[3]Department of Earth and Planetary Sciences, Yale University, New Haven, CT, USA

**Correspondence:** Christopher Horvat (horvat@brown.edu)

**Abstract.** Sea ice coverage is a key indicator of changes in the global climate. Estimates of sea ice area and extent are primarily derived from satellite measurements of surface microwave emissions, from which local sea ice concentration (SIC) is derived. Passive microwave (PM) satellite sensors remain the sole global product for understanding SIC variability, but may be sensitive to consistent biases. In part I we explored these in a multi-sensor intercomparison of optical, passive microwave, and lidar data, showing that a new, independent SIC product, the linear ice fraction (LIF), derived from ICESat-2 (IS2)laser altimetry, could be used to quantify and understand PM SIC biases. Here in part II, we develop and assess the reliability of larger-scale estimates of SIC from IS2 LIF. We develop an LIF emulator that samples optical imagery using the distribution of possible orientation angles for IS2 to understand the limitations of this one-dimensional product. We find that the error qualities of the LIF product are improved when combining multiple IS2 tracks, and discuss intrinsic but correctable biases that emerge in the combination of multiple IS2 measurements. We use these to develop a monthly LIF product, covering up to 54% of the Arctic sea ice cover, with has similar-or-better error qualities compared to PM data. We then discuss pathways to enhancing PM-SIC data with IS2 LIF in the future.

## 1 Introduction

Sea ice concentration (SIC), the fraction of an ocean area covered by sea ice, is critically important for understanding polar climate variability. SIC is estimated globally using passive microwave (PM) satellites at both hemispheres, with PM-derived SIC the standard for assessing sea ice state and change (Meredith et al., 2022). Increasingly, SIC products are assimilated into state-of-the-art forecast and climate models at both hemispheres (Mazloff et al., 2010; Sakov et al., 2012; Massonnet et al., 2015; Verdy and Mazloff, 2017; Fritzner et al., 2019; Zhang et al., 2021), making potential improvements in global SIC observations important for accurate climate analysis and prediction. Local errors in PM-SIC are observed to have a compensating effect when integrated over the Arctic or Antarctic, thus hence the impact of algorithmic uncertainty or bias on estimates of total (Arctic or Antarctic) sea ice area is estimated to be less than 1%, even in summer (Notz, 2015; Meier and Stewart, 2019; Kern et al., 2020). Still, no independent, unsupervised, alternatives to PM exists for measuring SIC from local to global scales.

In Part I of this two-part study (Buckley et al., 2024), we compared daily retrievals from state-of-the-art PM sensors and PM-SIC algorithms against high-resolution optical data from NASA's operation IceBridge. We calculated SIC from the optical



imagery by applying a surface type classification algorithm (Buckley et al., 2020) to the images, defining each pixel as open

water, sea ice, or melt pond, and determined a sea ice concentrtion for each  400 m by  600 m image. We found that PM-SIC

products demonstrated consistent positive biases (1-6%) over compact sea ice, potentially because of the presence of small

crack features in the sea ice mosaic that cover a limited portion of the overall surface and are challenging to capture with large

PM grid sizes (6.25 to 25 km cells). However, these fractures may contribute greatly to air-sea exchange. This intercomparison

showed a wide uncertainty range for PM-SIC summer months (May-September), because of the well-known challenges in

retrieval of SIC over ponded sea ice. Part I includes details of these biases and limitations of PM products.

We showed sea ice surface type retrievals from NASA's ICESat-2 satellite (IS2) can be used to develop a linear SIC estimate,

which we call the linear ice fraction (LIF), that has reduced or similar bias compared to PM over a set of imagery coincident

with IS2 overflights. IS2 is a photon-counting laser altimeter with 0.7 m along-track sampling, a 10-meter footprint, and high

skill in differentiating sea ice and open water in non-summer months (Farrell et al., 2020; Kwok et al., 2020, 2021). IS2 can

resolve Arctic leads at the meter scale (Petty et al., 2021; Kwok et al., 2021), especially in winter, when leads are the primary

source of air-sea exchange. Importantly, IS2 does not rely on the PM signature of sea ice or water and therefore has independent

uncertainties from PM-SIC. Yet these uncertainties are largely unconstrained and could potentially be much larger than PM

products.

Here we explore error bounds with IS2 LIF, and the possibility of using multiple consecutive IS2 passes to build a gridded LIF

product on monthly timescales. We first discuss the uncertainties that arise in an IS2-derived gridded product. To understand

them, we develop an IS2 emulator which we apply to the optically classified sea ice data explored in Buckley et al. (2024) in

Sec. 2.1, using it to derive bounds on how unsupervised errors in SIC retrieval decay as a function of the number of IS2 passes.

By using the error bounds obtained from emulation, in Sec. 3 we build a monthly Arctic LIF product that covers roughly 60%

of Arctic seasonal sea ice extent, and explore differences between it and a set of commonly-used PM-SIC products at different

resolutions. Over these areas, PM-SIC is approximately 3-4% higher in winter months, with LIF estimating approximately

twice as much open water than PM-SIC products, similar to what was obtained from optical comparisons. Finally, we explore

prospects for improving LIF skill, and how, either in single IS2 passes or as a gridded product, it could be used to augment

existing PM-SIC data in Sec. 4.

**2   ICESat-2 and the Linear Ice Fraction**

ICESat-2 (IS2) is a 6-beam laser altimeter with high precision and skill in retrieving sea ice properties (e.g Kwok et al., 2019a).

In this work, and in Buckley et al. (2024), we use the sea ice height product, ATL07, which generates along-satellite-track

"segments" from collections of sequential 150 photons. Based on the statistical properties of such photons retrievals, each

segment is identified with a surface type (water, ice, or cloud covered) (Kwok et al., 2019b). These segments are provided in

locations where the local daily NSIDC-CDR sea ice concentration exceeds 15% and their length averages ~15 m for the 3

strong beams and ~60 m for the 3 weak beams (Kwok et al., 2019a).





As detailed in Buckley et al. (2024), for any collection of measured IS2 segments, we define the IS2 linear ice fraction (LIF) as:

$$LIF = \frac{\text{length of ice segments}}{\text{length of all surface segments}}. \tag{1}$$

The details of the ATL07 segment type classification can be found in the Algorithm Theoretical Basis Document Kwok et al. (2019a) and we follow the preprocessing methods in Horvat et al. (2020b). We exclude all cloud segments, sections with fewer than two segments within 1 km along-track, and all segments over 200 m long. Although LIF is calculated with a high precision instrument and not subject to the passive microwave biases in SIC determination, we note other independent sources of uncertainty.

**U1: Classification uncertainty**  The construction of LIF relies upon the IS2 ATL07 classification of along-track segments of the ice-ocean surface as being ice or sea water. Uncertainty in this classification, which is higher in summer Tilling et al. (2020); Farrell et al. (2020), introduces the potential for systematic error in LIF calculations.

**U2: Orientation uncertainty**  The relative orientation of near-linear features in the sea ice mosaic is unknown with respect to the satellite path. While the local azimuth of the IS2 satellite is constrained as a function of latitude (see Supporting

Figure S1 and Sec. 2.1), the orientation of sea ice features is not. This can distort the fraction of the observed surface that is ice or open water if the alignment of cracks and overflights is correlated (Rothrock and Thorndike, 1984; Horvat et al., 2020a; Hell and Horvat, 2024).

**U3: Coverage uncertainty:**  PM satellites cover the entire Arctic approximately once per day and are not influenced by clouds. IS2, however, makes approximately 15 orbits each day, with all six beams spanning a region 25 km wide, and its photons

do not reach the sea ice surface through cloud. IS2 cannot be relied upon to produce specific measurements of the sea ice surface at any one location over the short repeat time of PM satellites, and therefore gridded products may be formed by averaging temporally intermittent IS2 samples.

Improving classification uncertainty (**U1**) is a significant area of ongoing research with IS2 (Petty et al., 2021). As it pertains to LIF, in Buckley et al. (2024), we explored **U1** by intercomparing IS2 overflights and PM-SIC measurements over four

coincident high-resolution optical images. The present classification scheme in ATL07 version 7 yields single-pass LIF ($LIF_0$) values similar or better in their estimation of SIC than PM-SIC products (overall XX%). The three beams from the single pass of ICESat-2 over an image produce a 2.4% bias, while the average PM bias over the same area is 3.75% Buckley et al. (2024). Yet even when IS2 classification is "perfect" (according to classification data from the optical imagery), error associated with **U2** from a single beam IS2 pass limits the best case error - in the selected imagery examined in Buckley et al. (2024), this was

approximatdly 1.0%. Because the orientation of IS2 overflights and crack features is a priori random, repeat measurements can help to reduce **U2** by sampling a broader variety of sea ice geometric variability. Yet sea ice motion can alter the makeup of sea ice in a region over the time period in which IS2 would return. A compromise is necessary between incorporating more repeat tracks and therefore reducing **U2** and the temporal resolution of any gridded product, while still accounting for the fact that



reducing the temporal resolution does not guarantee increased repeat tracks **U3**. The focus of our development of a gridded
LIF product will not be on eliminating classification uncertainty. We hope to produce a product that encompasses the largest
sea-ice-covered-area as possible while still minimizing error **U2**. To do this we build an IS2 emulator, which simulates IS2
passing over optically-classified sea ice, which we will use to investigate LIF error bounds as a function of overflight number
in Sec. 2.1. To address **U3**, we note that lower temporal resolution allows for more areas to be sampled sufficiently by IS2
although care must be taken to ensure that the underlying sea ice comparison is appropriate between PM-SIC samples and IS2
samples. We discuss the requirements of this product in Sec. 3.

## 2.1   Error bounds on IS2-SIC from emulation

To understand orientation uncertainty **U2**, we build an IS2 emulator, schematically shown in Fig. 3 over an example OIB
image. The emulator code is provided publicly at Horvat (2024b) (see Code and Data Availability). We describe the emulator
in detail below, but in summary, for each image we build a series of synthetic single-beam overflights that match the known
orientation of IS2 reference ground tracks (RGTs) at the image location. The surface is then intersected with a number of such
appropriately-oriented tracks, and LIF is calculated for each along-track intersection. We apply this technique to the full set
of 70,000+ optically-classified images described in Buckley et al. (2024). These images are 17,000 scenes from the operation
icebridge summer campaign in July 2016 and July 2017, and 53,000 scenes from the winter campaign in March and April
2018. Using this extensive dataset we can investigate how LIF error changes with the number of passes and latitude.

105       We first identify each optically-classified image with its corresponding latitude. The distribution of RGT azimuths (angles
with respect to local North) varies as function of latitude alone and is specified according to the IS2 91-day repeat cycle. Thus
at each latitude, we identify the distribution of possible RGT azimuths from the IS2 Technical Specifications (Neumann et al.,
2019), with the probability distribution shown in Fig. 1(a). We sample from this distribution at each latitude using inverse
transform sampling to obtain a distribution of RGT orientations for a Monte-Carlo-style emulation of the LIF computation. For
most latitudes, the RGT azimuth distribution has approximately only two possible directions (Fig. 1b), though because of the
increased track density, the distribution widens approaching the pole (compare the azimuth PDF at 87N (red) to 70N (black)).

Fig. 2 shows the application of the emulator to an image from the optically-classified dataset used in Buckley et al. (2024),
an image with a sea ice concentration of 92%. Given an azimuthal angle, we select a "tie point" in the image (red dots, a),
and draw a straight-line synthetic RGT (SRGT) through that tie point at the specified orientation angle (black lines). We then
compute the length of ice-covered points and ice-free points in the image that are intersected by the SRGT, storing them as a
function of each SRGT crossing (Fig. 2b, blue and grey lines). This process is repeated to develop a series of SRGTs distributed
according to the known IS2 RGT orientations for each image. We evaluate M=100 total crossings for each image, though in
practice any number is possible. For any SRGT, we can compute $LIF_0$ as the fraction of "overflown" ice points to overflown ice
or ocean points for any individual SRGT (red dots, Fig. 2c). Because the classified imagery is provided on an equal-area grid,
we simply count the number of ice and ocean points overflown by the synthetic RGT when evaluating the along-track lengths.
In application to real data (Eq. 1, and applied in Buckley et al. (2024), the LIF is computed by weighting each segment by its



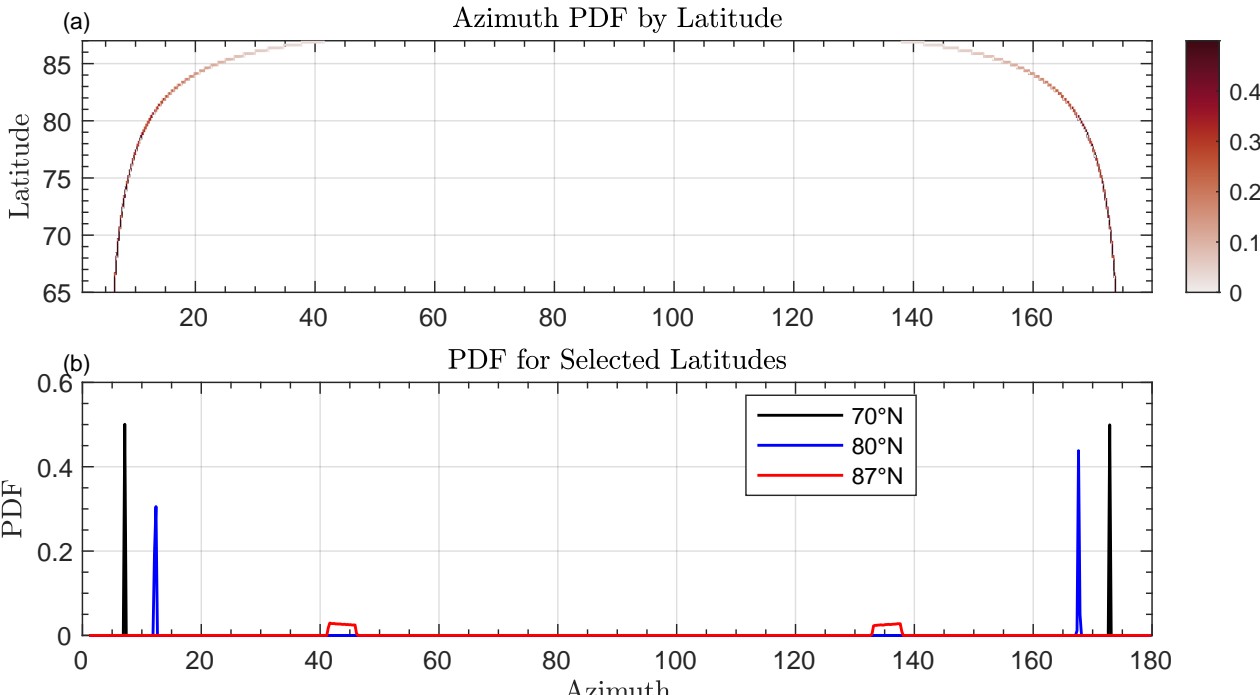

**Figure 1.** Direction of IS2 transit with respect to a line of longitude (satellite azimuth) as a function of latitude. (a) Probability distribution of azimuthal angle as a function of latitude for all Arctic IS2 RGTs. (b) Probability distribution for latitudes 70N (black), 80N (blue), or 87N (red).

length. To recreate the process of observing the LIF in reality, we evaluate $LIF_n$, which is the cumulative sum of ice points divided by the cumulative sum of all ice points for $n$ SRGTs (solid black line).

A priori, there is significant variability in $LIF_0$ measured by any one SRGT. Such single-pass error for individual images was what was examined in Buckley et al. (2024). For the example image (Fig. 2a), while the mean difference in $LIF_0$ from the true SIC across all 100 SRGTs is -1.7%, the standard deviation is $\pm$ 13.55%. These different $LIF_0$ measurements are scattered as red dots in (c). On pass 8 (red line, a), for example, the SRGT intersects almost entirely with a region of open water, recording an $LIF_0$ of just 75.1% (not shown in (c)). This high variance for single-passes is what necessitates an accumulation of multiple SRGTs in an gridded product. To facilitate an understanding of how many intersections might be required to obtain a suitable estimate of SIC, we first define an optimal LIF for each image, $LIF_i^*$, where

$$LIF_i^* \equiv \lim_{n \to \infty} LIF_n. \tag{2}$$

where $LIF_n$ is the LIF formed using $n$ consecutive intersections of image $i$. Because the orientation of SRGTs can be correlated to the geometric features of a surface, LIF* is not necessarily equal to the SIC. Thus we define the optimal bias, $B_i = LIF_i^* -$




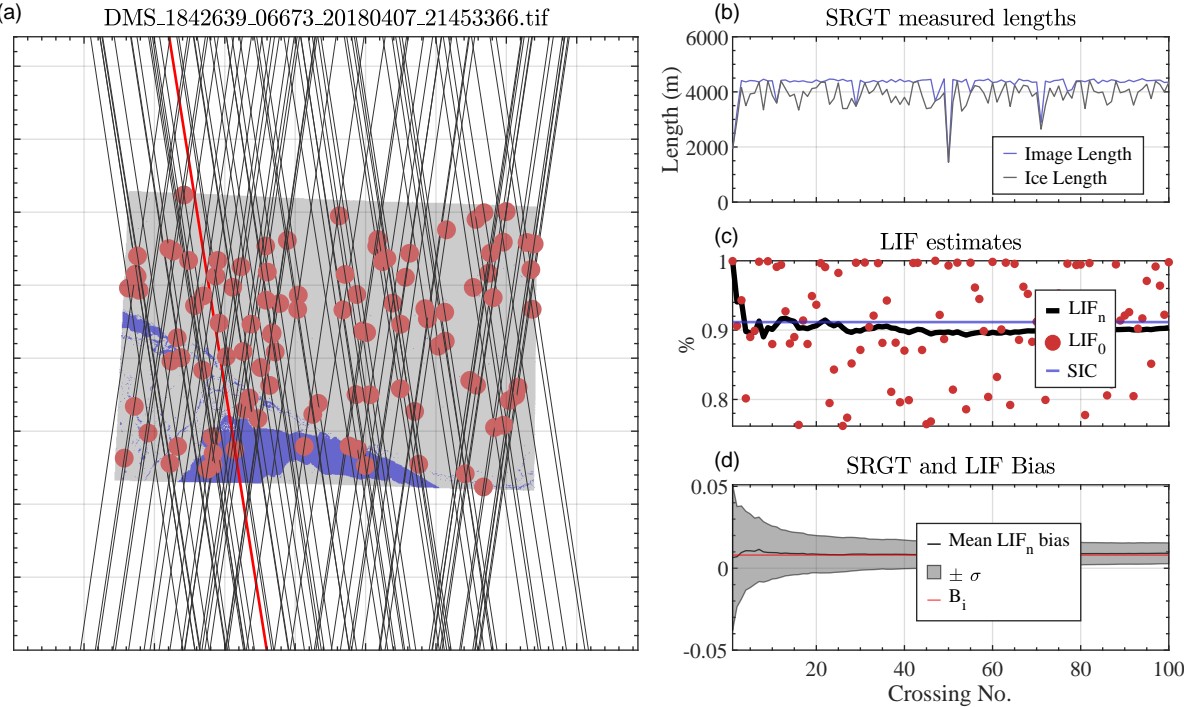

**Figure 2.** Example application of IS2 emulator to a classified DMS image from IceBridge. (a) Classified optical image from Operation Icebridge. Blue points are water, grey points are ice. Black lines are synthetic IS2 RGTs, which pass through randomly-generated tie points (red dots). The highest-bias SRGT is shown as a red line. (b) Total length of image sampled (blue) compared to ice points sampled (grey) for each SRGT crossing in (a). (c) Single-pass LIF ($LIF_0$) estimates from SRGT data in (b) (red dots), and $LIF_n$ derived from cumulatively integrating the SRGT passes in the order in (b) (solid black line) compared to true image SIC (blue horizontal line). (d) Mean $LIF_n$ bias from true SIC (black line) and standard deviation (shaded region) across all permutations of SRGTs in (a).

$SIC$, which is the best-case **U2** error in LIF for each individual image. Below (see Fig. 3) we show $B_i$ is tyically small with a near-zero mean across all classified imagery.

The progression from $LIF_0$ to $B_i$ is path-dependent. While any suitably dense set of SRGTs will approach $LIF^*$, this progression is not monotonic. For example, accumulating across the SRGTs ordered as in (c, black line) yields an estimate $LIF_8$ that is 2.2% larger than the true SIC, although the difference from the true SIC is less than 1.9% for $n > 8$ and $B_i$ is just 0.8%. Altering the order in which the SRGTs are accumulated can yield a faster or slower progression towards $B_i$. We can make use of the emulator to constrain two components of the overall uncertainty **U2**:

**U2a: Image Uncertainty** Associated with differences between image geometries on convergence to $B_i$.

**U2b: Sampling Uncertainty** Associated with the path-dependent convergence to $B_i$.





As SRGTs are randomly drawn, any permutation of SRGTs is equally likely and both the expected error at crossing $n$, and the variability in that error, can be quantified by exploring a range of paths to $B_i$. For each image, $i$, we generate a large set

of estimators of $LIF_{i,n,k}$, where $k$ is any of the possible orderings of the $M$ SRGTs of length $N$, sampled with replacement. Sampling without replacement leads to a convergence of each permutation to $B_i$ as all SRGTs are accumulated. By sampling with replacement, we form a bootstrap estimate of **U2b**, and one that can be deployed in an operational context, where the number of RGT intersections will be limited. We define,

$$E_{i,n,k} \equiv LIF_{i,n,k} - SIC_i, \tag{3}$$

the error in SIC for image $i$ after accumulating $n$ crossings under SRGT ordering $k$. For each image, we sample $M = 100$ SRGTs from the distribution of RGTs at the image latitude (Figure 1b). From these, we generate $P = 400$ ordered LIF estimates of length $N = 100$ by sampling with replacement from the SRGTs. For each image, then, we have $P \times N = 40,000$ representations of the LIF. Defining a mean over permutations by an overline, Figure 3(d) shows the evolution of $\overline{E}_{i,n}$ for the example scene in (a), as well as the standard deviation $\pm S_{i,n}$ (filled lines) computed across the bootstrapped samples at each

intersection number, which asymptotes to a value $S_i$. For the image in (a), the rapid convergence of $\overline{E}_{i,n}$ to $B_i$ is expected given that $\overline{E}_{i,1}$ is the mean of $P$ $LIF_0$ measurements. There is a slower convergence of $S_{i,n}$ to a value of $S_i = 2.1\%$ - the uncertainty **U2b** associated with sampling variability for this image.

We next explore the statistics of $B_i$ and $\overline{E}_{i,n}$, as well as the sampling uncertainty $S_{i,n}$ across the set of all images. In Fig. 3(a,b) we show the distribution of $B_i$ and $S_i$ for all images considered here, with interquartile ranges as solid black

vertical lines. For purposes of calculating $S_i$, we exclude imagery where the true SIC is less than 10% or greater than 99%. The mean long-term bias $B_i$ is -0.07 $\pm$ 1.4%, indicating that this method is suitable for capturing the large-scale statistics of SIC to reasonable error, if suitably many RGTs intersect a region. The sampling uncertainty $S_i$ has a mean of 1.27%, and 95% of $S_i$ values are less than 1.77%.

We show the two senses of uncertainty comprising **U2** in Fig. 3(b). The solid line plots $\langle E \rangle_n$, which has a near-zero mean,

even for $n = 1$. Superimposed on $\langle E \rangle_n$ are two cones of uncertainty. The first, in grey lines, shows a range of $\pm \langle S \rangle_n$, or the mean sampling uncertainty at each crossing number across all images. This is, as in Fig. 3, high for several initial crossings before decaying and asymptoting to the long-term mean of 1.27% indicated above. In blue lines we plot the standard deviation of $\overline{E}_{i,n}$, the bias removing the uncertainty associated with sampling variability. Here, this value is approximately constant at the long-term mean of $S_i$, as each value of $\overline{E}_{i,n}$ is an average of many LIF measurements which should represent the standard

error. Thus we see that in general, uncertainty associated with **U2** can be characterized in terms of an intrinsic image-based uncertainty $\langle S \rangle$ plus a path-dependent uncertainty that declines with increasing cross-number. The number of intersections we choose to require when building an unsupervised product can be varied as required. For example, to reduce the uncertainty below a threshold of 2.5% requires more than 7 crossings. Since we use only single-beam estimates, and the IS2 altimeter has 3 strong beams and 3 weak beams, these 7 crossings could equally represent 2 or 3 IS2 overflights, depending on whether



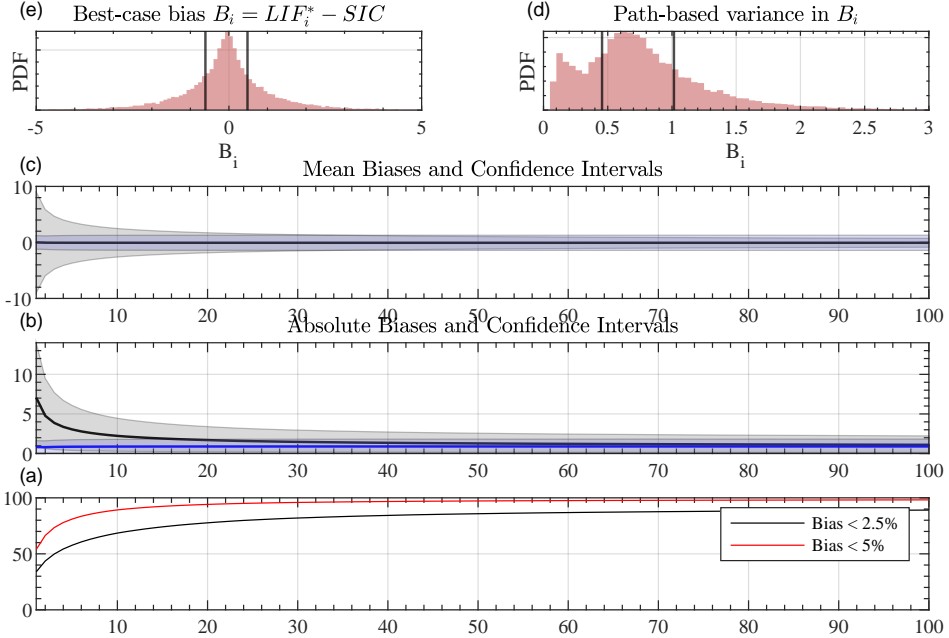

**Figure 3.** Statistics of SIC reconstruction with LIF using data from all 70,225 classified Operation Icebridge images. (a) Histogram of best-case bias in SIC, $LIF_i^* - SIC$. Vertical lines are interquartile range of -0.60% to 0.49%. (b) Histogram of variance in bias, $S_i$. Vertical lines are interquartile range of 0.46% to 1.02%. (c) Mean bias (black line) as a function of intersection number. Interquartile ranges are (blue) treating all images and path permutations separately or (black) averaging permutations for each image first. (d) Same as (c) but for the mean bias. Vertical line shows the 2.5% mean absolute error cutoff.

weak beam measurements are included. In the application below, we require 8 total beam intersections when building an unsupervised monthly product to an error tolerance of 2.5%.

## 3 A Global ICESat-2-based LIF Product

Above, we showed that a sequential LIF-style approach can be used to reconstruct the surface sea ice concentration with high precision and well-constrained error statistics, although important uncertainties remain (discussed in Sec. 2.1 and below). Leveraging this uncertainty information, we build an IS2-based SIC product. Here we analyze only the Arctic product, but we provide Antarctic LIF data in Horvat (2024a). These data and code for generating a global gridded product of LIF-based SIC are provided through the MATLAB-based package IS2-Grid version 0.4 (Horvat, 2024a). This software package is designed to produce modular gridded sea-ice-related products at requested temporal and spatial gridding through an accumulation of multiple tracks, for comparison with climate model and observational data. It permits the rapid development of cumulative statistics over chosen temporal windows, and currently provides estimates of the floe size distribution, significant wave height,



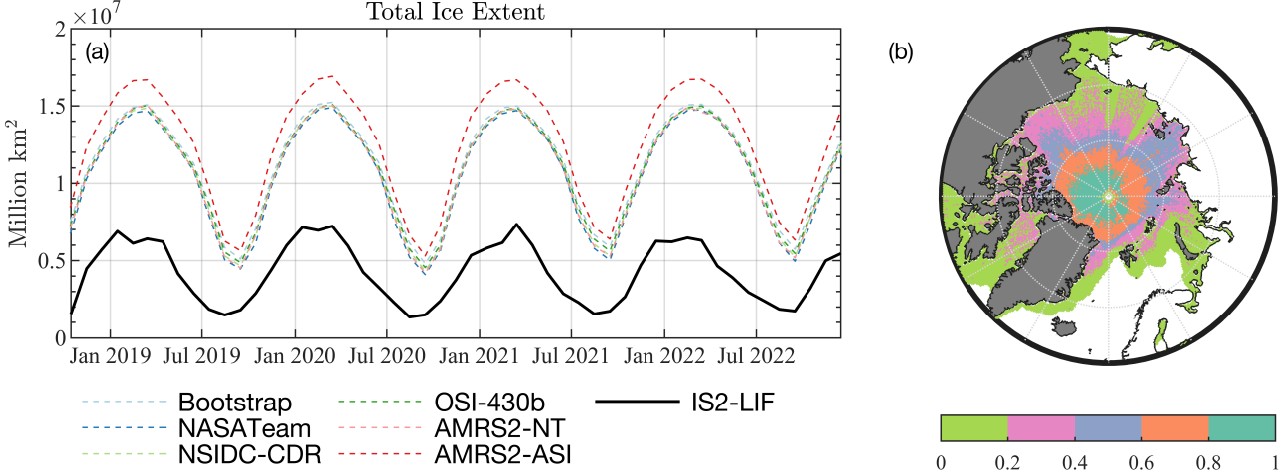

**Figure 4.** Comparison of coverage of IS2 LIF data to commonly-used PM-SIC products. (a) Arctic sea ice extent of 6 PM-SIC products (dashed colored lines) compared to the area well-sampled by IS2 (black line) from October 2018-December 2022. (b) Percentage of months from October 2018-December 2022 where PM-SIC record sea ice and IS2 tracks are sufficiently dense.

and sequential LIF along with other ancillary statistics. This code is modular and provides a simple way for creating gridded products from along-track-calculated statistics. Here we examine an LIF product using this code base, which generates a monthly LIF product on the 25km polar stereographic grid, which is the same resolution of most target PM-SIC products (see Buckley et al. (2024)).

## 3.1 Tradeoffs in temporal sampling

In addition to the uncertainties with orientation and surface classification, when building a longer-time-scale product, we must consider that IS2 overflights exhibit temporal intermittency compared to PM measurements that are retrieved daily. At each grid point, we define an "ICESat-2 intermittent" PM-SIC, $\tilde{c}$, equal to the segment-averaged PM sea ice concentration using the along-track defined PM-SIC. Two reference PM datasets are included along-track with the IS2 ATL07 product, the NSIDC CDR (all ATL07 versions) and the AMSR2-NT product (ATL06 v6 and later). We define the "temporal intermittency bias", $B_T$ from a monthly-average SIC, $\bar{c}$, as,

$$B_T = \tilde{c} - \bar{c}. \qquad (4)$$

The value of $B_T$ is a measure of how different the PM-SIC product would be if sampled only when IS2 flew overhead from the true PM-SIC monthly mean, and estimates the bias introduced by IS2's intermittent temporal sampling. Conscious of the impact of examining a time-evolving SIC surface, we ignore any grid cell where $\|B_T\|$ exceeds 5%, defined using the NSIDC CDR product. This reduces the number of grid cells over which we develop an LIF product. We combine this restriction with the requirement of 4 or more IS2 crossings discussed in Sec. 2.1. We also require that all SIC estimates report greater than 15%





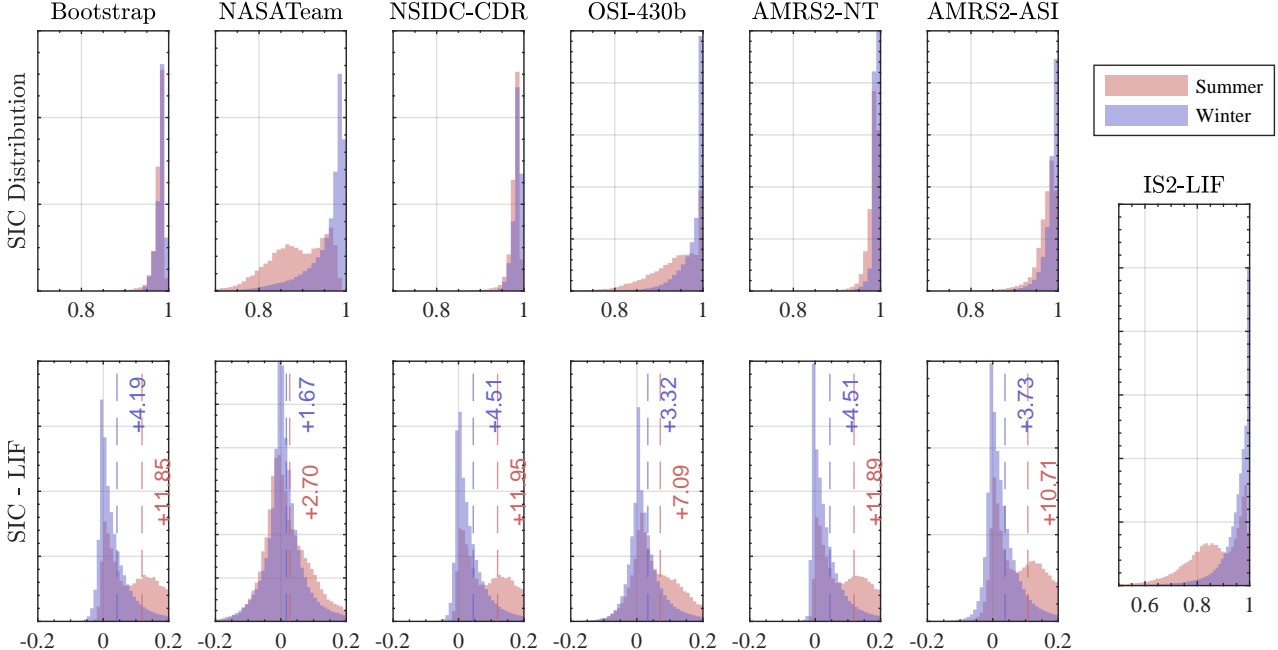

**Figure 5.** Histograms of PM-SIC for 6 products (top row) and the difference from IS2 LIF (bottom row). Summary statistics are provided in Table 1. Red colors are summer months, blue are winter months. Vertical lines and labels indicated median $\Delta$ values.

SIC at a given location. Figure 4(a) shows total sea ice extent for all PM products (dashed lines), compared to the area over which we can make an IS2-PM comparison (solid line). There are data for nearly all months above 80°N, as a consequence

of the higher track density near the pole, but this coverage declines with latitude. This dense coverage in the central Arctic is particularly useful for quantifying lead detection by PM in areas of high SIC. We show in (b) the fraction of months during the IS2 operational period (here from October 2018-December 2022) where there is sea ice recorded by PM-SIC and sufficiently dense IS2 tracks. In any given month the area with compatible IS2 coverage for comparison with PM-SIC is between 24% and 54% (Fig. 4a) of Arctic sea ice extent.

**3.2 Comparison of gridded LIF data with passive microwave products**

For a global comparison with PM-SIC, we evaluate the monthly 25km LIF dataset against 6 widely-used PM-SIC products. Figure 4 shows the histogram of SIC values (top row) and histogram of differences from LIF (bottom row) for all data for each product, with LIF values offset right. We segment data into "summer" data from June to September (red), and "non-summer" data covering October to May (blue). Statistics derived from these distributions is given in Table 1, along with interquartile

ranges and median differences from LIF (shown using the symbol $\tilde{\Delta}$. In total, there are approximately 41,000 "summer" comparison points, covering 27 million km$^2$, and 290,000 "non-summer" comparison points, covering 189 million km$^2$ - larger



| Period | | "Summer" (Jun-Aug) | | "Winter" (Sep-May) | |
|---|---|---|---|---|---|
| Number | | $41 \times 10^3$, | | $290 \times 10^3$, | |
| Area | | $27 \times 10^6$ km$^2$ | | $189 \times 10^6$ km$^2$ | |
| | | $\overline{\text{SIC}}$ | $\tilde{\Delta}$ (25%,75%) | $\overline{\text{SIC}}$ | $\tilde{\Delta}$ (25%,75%) |
| Product | LIF | 86.2% | $\emptyset$ | 94.3 | $\emptyset$ |
| | LIF (specular leads) | 92.5% | 3.1% (0.8,9.1) | 94.7% | 0.0% (0.0,0.2) |
| | Bootstrap | 98.1% | 10.3 (2.8, 18.0) | 92.5% | 1.8% (-0.2,5.6) |
| | NASATeam | 88.9% | 0.9% (-2.5,6.7) | 96.0% | 0.5% (-1.9, 3.8) |
| | NSIDC-CDR | 98.2 | 10.4% (3.0,18.0) | 98.8% | 2.1% (0.1, 5.8) |
| | OSI-SAF | 93.3% | 4.8% (1.0,11.9) | 97.6 | 2.1% (-0.4,4.9) |
| | AMSR2-NT | 98.1% | 10.2% (3.0,17.8) | 98.8% | 2.1% (0.0,5.8) |
| | AMSR2-ASI | 96.9% | 9.0% (1.9,16.7) | 98.1% | 1.6% (-0.4,5.4) |

**Table 1.** Comparison of "summer" (May-Sep) and "winter" (all other months) statistics of IS2 global LIF product and the set of 6 examined PM-SIC products. $\Delta$ values are differences from standard LIF product. $\tilde{\Delta}$ is median difference, and values in parentheses the interquartile range of $\Delta$ (25%-75% intervals).

because of the larger spatial extent of sea ice and greater number of months included. We see that PM-SIC products indicate a higher ice fraction than the LIF in all seasons. Wintertime biases are similar to that found in OIB data as well as in classified optical data, with a median positive difference of 0.5-2.1% for sea ice that recorded by LIF as being 94.3% ice-covered on
average. Considering only specular returns as leads resulted in a better agreement between LIF and PM-SIC products in both seasons. This presents to the possibility that non-specular or "dark lead" returns obtained by IS2 are open water areas missed by passive microwave satellites, or also thin sea ice that is misclassified by IS2.

Summer $\Delta$ values are more variable, and with known issues in classifying surface meltwater in both PM and IS2 products (Kwok et al., 2019b; Tilling et al., 2020; Buckley et al., 2023; Herzfeld et al., 2023), caution should be exercised in the
application of either in these months. In general we find a positively-skewed distribution of summer values in the PM products which ranges from 0.9% to 10.4% on average. This overestimation is considerably more sensitive to the inclusion or exclusion of "dark" leads from the ATL07 product, which account for approximately 6.3% of sea ice in usable grid cells in these months. Interestingly, LIF and NASATeam PM-SIC algorithms are most similar, with the smallest mean difference and an interquartile range that includes zero, as in winter months. Using only summer leads classified as specular resulted in similar distributions
of LIF values to those from the PM-SIC products, though errors in the retrieval of summer sea ice properties from IS2 should continue to be the object of future study, to establish whether PM-SIC products are overestimating SIC in the melt season on average or not.





## 4  Conclusions

In this study, we developed a new gridded data product from the ICESat-2 laser altimeter, which we used to represent monthly
maps of sea ice concentration. We evaluated errors in the representation of the sea ice surface using an emulator which is run
on a set of classified optical images from NASA's Operation IceBridge. We showed that, in general, PM-SIC measurements
were positively biased against IS2 estimates, particularly in winter, as was the case when compared to imagery in Buckley et al.
(2024) and in previous literature (Kern et al., 2019). IS2 is particularly effective at estimating SIC, even with a limited number
of overflights, especially in regions of compact sea ice with leads. With further validation of the ATL07 surface classification
scheme, this product may help reduce open water biases significantly.

The IS2 linear ice fraction (LIF product is provided as a global, monthly product covering 25-54% of the Arctic sea ice
zone. This data product is available through December 2024 (see Data Availability). Because of the available comparative data
from Operation IceBridge, we only included Arctic comparisons in this work, though the data product is available in both
hemispheres. In months from October-May ("winter"), we found that the offset between LIF data and PM-SIC product data
was of the same order of the bias between the OIB optically classified imagery and PM-SIC data we found in Buckley et al.
(2024). Because of this consistency, we suggest that this captures an overestimation bias in the PM-SIC products, and this
offset is not from misclassification error in the ATL07 product.

In summer, lower LIF values compared to PM-SIC contrasted with expectations for heavily ponded sea ice in OIB imagery,
where we found SSMI/S-based PM-SIC products underestimated SIC Buckley et al. (2024). As in winter, the classification of
specular returns alone as open water resulted in a similar distribution of SIC values as PM-based products. We found that the
LIF product was most comparable to the NASATeam algorithm applied to SSMI/S data in these months, a data product which
had the largest biases compared to "ground-truth" OIB imagery in summer. The current IS2 algorithm does not distinguish
between melt ponds and open water, and the LIF product is likely not yet capable of enhancing PM-SIC products in these
months. Further work on the classification of ponded surfaces is needed before using a summer LIF-based SIC product.

As it illuminates biases, particularly in compact sea ice in winter, LIF derived from IS2 offers an independent and unique
opportunity to enhance estimates of sea ice concentration. Underestimations of SIC in the wintertime Arctic may be small:
but these differences correspond to large increases in open water fraction, which can drive ocean and atmospheric variability.
Climate models that are tuned to reproduce SIA from PM satellites, or that assimilate PM-SIC for forecasts, may underestimate
the magnitude of this air-sea exchange. We have provided validative data for LIF by using high-resolution optical imagery and
an emulation tool. It will be necessary to enrich this LIF data with more constraints to ascertain the year-round and repeat
skill of LIF and its potential for developing a new SIC data product on shorter timescales. IS2 offers a high-resolution and
repeatable opportunity to provide improved PM-SIC measurements and greater understanding of overall sea ice variability in
the polar seas.



*Data availability.* The monthly LIF product, and statistics from operation icebridge and worldview imagery will be provided as a Zenodo
repository upon paper acceptance. A release of the IS2 emulator is archived at Horvat (2024b) and accessible at github.com/antipodalclimate/IS2-
Emulator. A release of the IS2 gridded product generation code is archived at Horvat (2024a) and accessible at github.com/antipodalclimate/IS2-
Gridded-Products. Code to reproduce paper figures and statistics is available at https://github.com/antipodalclimate/IS2-LIF-paper-2024.

*Author contributions.* CH conceived of and developed the LIF product, performed the IS2 analysis, and wrote the paper. EB developed the
classification algorithms,provided OIB data and analysis of PM bias. MS and PY developed the IS2 emulator and tested its applicability on
data. All authors consulted on the scientific approach and content and EB and MMW contributed to writing the paper.

*Competing interests.* The authors declare no competing interests.

*Financial support.* This research was supported by NASA (80NSSC20K0959, 80NSSC23K0935, 80NSSC23K0782), NSF (2146889) and
Schmidt Futures—a philanthropic initiative that seeks to improve societal outcomes through the development of emerging science and
technologies.





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
