# Peer review of "Sea Ice Concentration Estimates from ICESat-2 Linear Ice Fraction. Part 2: Gridded Data Comparison and Bias Estimation."

_EGUsphere, 2024_

## Author Response (AR1)

Dear Dr. Howell,

Below we respond to both reviewers' comments in a single response document. We very much enjoyed the process of improving the LIF product, especially a detailed analysis of dark and specular leads and intra-beam variability. Through this process, we found differences in weak and strong beam data, shown in the new Figure 6, added additional supporting Figures to the manuscript to explore the sensitivity of our results to changes in processing, and switched to excluding dark leads from the product produced as a result of this analysis.

In general, we made efforts to address our presentation of passive microwave returns (generally responsive to reviewer 1), and the scope of the LIF product (generally responsive to reviewer 2). Both highlighted the need to clarify our section on error evaluation and the emulation scheme, and we have made an effort to do so in the revised manuscript. Reviewer comments are here reproduced in blue font and our responses in black. Changes made in the text are in red font and in indented quotes with the respective line numbers.

Best,

Christopher Horvat on behalf of the authors.

**Reviewer 1**

This paper presents a method to estimate gridded sea ice concentrations (SIC) from a novel Linear Ice Fraction (LIF) product derived using an emulator of ICESat-2 tracks. The paper is a companion to part 1 which compares the LIF product to passive microwave SIC products. Here in part 2, the authors examine and quantify the potential sources of bias when using a high-resolution linear product with gridded, low-resolution passive microwave SIC observations. The results look promising and the paper highlights the potential for these linear observations to enhance current passive microwave SIC products. I think this paper needs some revision before it is ready for publication, specifically to improve clarity of Section 2.1. There are also quite a few subtleties about passive microwave sea ice data that are not quite captured in the current paper. My comments below describe these issues in more detail.

**Comments:**

• L5: ATL07 relies on a passive microwave SIC product and a SIC threshold, thus, LIF is not really an independent SIC estimate.

ATL07 data are made available only where PM-SIC from NSIDC shows SIC greater than 15%. This sets an outer perimeter for when ATL07 products are available. However, once ATL07 segments are retrieved, LIF is computed only from 532nm photon heights and retrieval rates along-track and does not rely on microwave emissions themselves. We have clarified this as follows:

The geographic extent of LIF can depend on PM-SIC because ATL07 segments are only produced in regions where SIC from the NSIDC Climate Data Record (Meier et al., 2021) exceeds 15%. LIF itself is derived exclusively from the classification of IS2 photon returns (at a wavelength of 532nm) and does not rely on microwave emissions (wavelengths on the order of 1 cm) or related algorithms, and therefore has independent and separate uncertainties from PM-SIC. Such uncertainties are

presently unconstrained, and thus potentially larger than PM-SIC products, the focus of this work.

Additionally all areas suitable for intercomparison have SIC above 15% in the NSIDC CDR by design, which was something highlighted in the quality control steps. We elaborate this point slightly now:

We also require that all PM-SIC estimates report greater than 15% SIC at any location, which eliminates any potential dependency of intercompared LIF data on the PM-SIC 15% cutoff used to define ATL07.

The vast majority of intercompared points are in compact ice zones, meaning the possibility that the imposition of this extent mask affects presented results is highly limited. We note this in the revised manuscript:

The evaluation of LIF in representing local SIC primarily focused on areas of compact sea ice in Buckley et al. (2024), and because of the preprocessing steps employed in generating the monthly LIF product, nearly all locations of intercomparison in Sec.3 were also compact ice. For example, as indicated in Table 1, mean NSIDC-CDR in the intercompared regions for producing Figure 5 exceeds 98% - just 0.07% of points had an NSIDC-CDR less than 80%. This limits the degree to which the 15% NSIDC-CDR mask used to define the ATL07 can influence LIF data. The LIF product has not yet been validated for low-concentration ice or the CDR-defined marginal ice zones, and its utility in those regions remains an open question, although these areas are critical for understanding overall sea ice variability (Bennetts et al., 2022; Squire, 2022; Horvat, 2022).

• L14: Although they both measure the amount of area covered by ice, sea ice concentration (expressed as a percentage) and sea ice fraction (expressed from 0 to 1) are different. The two are used interchangeably and imprecisely throughout the paper. Here (L14), I would change "fraction" to "percentage" since you are defining concentration but consider if the language throughout the paper should be revised to use only sea ice fraction to simplify the discussion since this is how you defined your LIF product.

To address this ambiguity in definition, we report all LIF values and biases in units of percentage rather than in decimal notation. See revised Figures and text, e.g.

For any collection of measured IS2 segments, we define the IS2 linear ice fraction (LIF) as:

$$LIF = 100 \times \frac{\text{length of ice segments}}{\text{length of all surface segments}}.$$
 (1)

We represent the LIF as a percentage for consistency with typical usage of SIC data.

• L55: This applies in other parts of the paper as well, but the SIC products used in this paper need to be defined (e.g., acronyms spelled out) and cited appropriately. I see this is included in part 1, but it needs to be in part 2 as well.

Indeed, our citing was insufficient here. We added a paragraph to this end as follows:

This monthly 25km LIF dataset is evaluated against 6 widely-used PM-SIC products. Four rely on brightness temperatures from the Special Sensor Microwave - Imager/Sounders (SSMI/S) onboard US Defense Meteorological Satellite Program flight units 16-18. They are (1) the NASATeam (NT) (Cavalieri et al., 1984) and (2) Bootstrap (BT) algorithms (Comiso and Sullivan, 1986), the (3) NSIDC Climate Data Record (CDR), equal to the maximum of the Bootstrap and NASATeam algorithms (Meier et al., 2014), and (4) the OSISAF Global Sea Ice Concentration climate data record (OSI450-a, up to 12/31/2020) and interim climate data record (OSI430-a, up to 2023) (Lavergne et al., 2019). We also include two algorithms using brightness temperature data from the Advanced Microwave Scanning Radiometer 2 (AMSR2) sensor on board the JAXA GCOM-W satellite, either (5) the NASAteam2 algorithm (Meier, 2018), or (6) the ASI-ARTIST algorithm Spreen et al. (2008). (1-3,5) are provided on the NSIDC 25km polar stereographic grid. We use OSI450/430 products (4) on the 25km EASE grid and (6) the ASI-ARTIST product on a 6.25km polar stereographic grid, both of which we regrid to the NSIDC 25km polar stereographic grid. We analyze PM-SIC and IS2 data across the time period from the launch of the IS2 satellite in October 2018 until December 2023 for 63 months including 5 full calendar years. Further details on the PM-SIC algorithms and satellite platforms used can be found in Buckley et al. (2024).

• L63: Similar comment as above, there is risk of passive microwave biases in your product because the data are not independent.

**See the above clarification and edit here:**

The geographic extent of LIF can depend on PM-SIC because ATL07 segments are only produced in regions where SIC from the NSIDC Climate Data Record (Meier et al., 2021) exceeds 15%. LIF itself is derived exclusively from the classification of IS2 photon returns (at a wavelength of 532nm) and does not rely on microwave emissions (wavelengths on the order of 1 cm) or related algorithms, and therefore has independent and separate uncertainties from PM-SIC. Such uncertainties are presently unconstrained, and thus potentially larger than PM-SIC products, the focus of this work.

• L73: I disagree here. 1) A PM satellite covers the entire Arctic approximately twice per day. 2) PM observations are influenced by cloud cover, especially liquid clouds or clouds with heavy precipitation. I think what you are meaning to point out here is that PM SIC observations can be obtained daily even in cloudy conditions.

Thank you for this correction. We made this sentence simpler as well:

PM satellite products yield daily SIC observations, even in cloudy conditions.

• L76-77: This is also true for most PM SIC products as well. They are usually produced from drop-in-the-bucket daily- or twice-daily- averaged brightness temperatures.

We now clarify to illustrate the temporal averaging of IS2 must happen on longer timescales:

IS2 cannot produce specific measurements of the sea ice surface at any one location at the daily or twice-daily repeat time of PM satellites. Gridded products can only therefore be formed by averaging temporally intermittent IS2 samples over longer periods than the daily or twice-daily PM repeat timescale.

• L80: Is ATL07 version 7 available somewhere or is there a reference to this information? I see release 6 is currently available at NSIDC.

This was a typo and we no longer allude to a future version.

The present classification scheme in ATL07 version 6 yields single-pass LIF (LIF $_1$ ) ...

• Fig 2c: Related to my comment above about concentration versus fraction, the unit for ice fraction is not "%" as labeled on the y-axis. You also have legend labels for LIF and SIC for data that is plotted only as a 0-1 range.

We modified Figure 2 and references to LIF to be in units of percentage now.

We represent the LIF as a percentage for consistency with typical usage of SIC data.

• Figure 3 and L136-176: I understand what you are accomplishing with the image and sampling uncertainty estimation and think it is a reasonable approach; however, there are many problems with the labeling on Figure 3 that make it extremely hard to follow this section of the text. I'm having a hard time parsing which parts of the figure the text is referring to because of the errors on the figure, in the caption, and in the text. This section and figure need a careful edit before they make any sense.

Indeed! This was highlighted by both reviewers. The figure itself has been checked for consistency with the text and appropriate labeling, and generally simplified and reproduced. See the new Figure 3 and the overall re-written Section 2.1, where we are more careful in our use of bootstrap replicates and in outlining the statistics we employ to produce Figure 3.

See Section 2.1, 2.2, and new Figure 3.

• L202: "4 or more IS2 crossings" – Does this mean overflights or beams? L174 mentions 2 or 3 IS2 overflights. Does this mesh with L202? Please clarify.

Indeed this was confusingly worded throughout. This should have read 8 crossings - though we now modify this to 11 thanks to changes in the analysis. Throughout we have identified all locations where "crossing", "ground track", "overflight", or related terms were used confusingly and attempted to rectify. We now clarify the distinction:

Here a "crossing" refers to the independent sampling of the sea ice surface by one IS2 beam, whereas an "overflight" refers to a general sampling of the surface by the IS2 satellite - this could lead to as many as 6 "crossings" by the 6 weak and strong beams.

• L228-229: I don't see any discussion of why the LIF and NASA Team are both bimodal. Why might that be? Or conversely, why are the other products not bimodal?

This is a great question - and spurred additional analysis in the manuscript, which we now explore through Figures 5 and 6, Supporting Figure S1, and Section 3.2. We now discuss explicitly this bimodality in Summer months,

The impact of dark lead segments on the overall LIF distribution can be seen in Fig.5, where the shape of the LIF histogram including all dark leads in summer months (gold histogram) is peaked at 81%, with no areas of 100% LIF. On average, including dark leads as open water leads to a reduction in LIF by 9.7% in July and August. . . .

We show the variability in this peaked nature in summer and discuss it,

The peaked distribution of LIF including dark leads contrasts with the histogram of LIF values in all other months (see "non-summer" months in Fig.4 and Supporting Figures S1 and S2), where the histogram of LIF values increases monotonically with LIF. This points to a role of surface melt in altering the surface returns and possible misidentification of "dark lead" segments. . . .

**And from this analysis suggest using the specular leads only in all months**

Because of the potential errors associated with dark lead classification, the similarity in SIC histograms with other PM-SIC products that are known to be biased or more uncertainty in summer months, and the minimal impact of dark leads outside of months with surface melting, we provide here as the core LIF product the one that includes only "specular" leads as open water in all months, which we denote LIFspec. . . .

• Table 1: The months noted for "summer" and "winter" periods do not match with the text (L214 and 244). Which is correct?

We fixed this typo as "summer" was taken to be June-August. See Table 1 and text changes, e.g:

To differentiate between these potentially melt-affected results, we segment the LIF data into "summer" or pond-affected months covering July and August (Fig.5 red and gold), and "non-summer" data covering September to June (Fig.5, blue).

• L255: As above, LIF is not quite independent.

See the above discussion:

The geographic extent of LIF can depend on PM-SIC because ATL07 segments are only produced in regions where SIC from the NSIDC Climate Data Record (Meier et al., 2021) exceeds 15%. LIF itself is derived exclusively from the classification of IS2 photon returns (at a wavelength of 532nm) and does not rely on microwave emissions (wavelengths on the order of 1 cm) or related algorithms, and therefore has independent and separate uncertainties from PM-SIC. Such uncertainties are presently unconstrained, and thus potentially larger than PM-SIC products, the focus of this work.

**Technical/typographical comments**

• Operation IceBridge appears in a variety of different permutations throughout the paper. Please use consistent capitalization and/or just use OIB after defining it at the first use.

Thanks - we have revised as suggested and OIB appears throughout after the first definition.

• L11: "with has similar" to "which has similar"

Fixed!

• L20: "thus hence", delete one.

Fixed!

• L26: typo – concentration

Fixed!

• L52: Which version of ATL07 was used?

We amended to:

In this work, and in Buckley et al. (2024), we use Version 6 of the sea ice height product, ATL07, which generates along-satellite-track "segments" from collections of sequential 150 photons (Kwok et al., 2023).

• L70: Where is Supporting Figure S1?

This was a typo, and should have referred to Figure 1 which was originally a supporting Figure.

• L81: XX% - fix

Fixed!

The present classification scheme in ATL07 version 6 yields single-pass LIF (LIF1) values similar or better in their estimation of SIC than PM-SIC products - with a single overflight of ICESat-2 over an image leading to an average 2.4% bias, with PM-SIC biases over the same areas of 2.9% or greater, and averaging 3.8% Buckley et al. (2024).

• L97: Should the in text figure reference here be to Fig. 2?

Yes! Fixed

• L137: "(c, black line)" which figure? Fixed!

• L155: Is this referring to panel a in Figure 2? Indeed, fixed!

• Figure 4 and L195: These SIC products need to be defined in this paper. See the revised referencing to other PM products above.

• Figure 5: In the caption and legend – should it be non-summer instead of winter? We have fixed the referencing to "winter" throughout to "non-summer".

• L225-226: Add an in text reference to Table 1.

We did so!

Summary statistics are provided in Table 1.

and

We compute summary statistics of LIF data evaluated using strong and weak beams alone in Table 1.

• L234: Use "IS2" to be consistent throughout the paper.

Fixed!

• L251: Define SSMI/S.

We edited this:

Four rely on brightness temperatures from the Special Sensor Microwave - Imager/Sounders (SSMI/S) onboard US Defense Meteorological Satellite Program flight units 16-18.

**Reviewer 2**

This paper introduces a new gridded dataset for linear lead fraction (LIF) derived from NASA's ICESat-2 laser altimeter. The study is a continuation of an earlier submission to The Cryosphere, now split into two parts based on prior feedback. The first part focused on OIB imagery classifications and the introduction of the LIF variable, along with initial comparisons with visual imagery. This second part presents a novel satellite emulator to analyze the impact of altimetry crossings on LIF estimates, followed by monthly gridded Arctic LIF estimates and their comparison to other sea ice concentration (SIC) products. I have reviewed both parts.

From a scientific perspective, the main difference between this study and the previous submission is the use of more realistic crossing angles (azimuths) in the crossings/emulation analysis, as well as a more detailed discussion of resulting biases. The methodology for generating monthly gridded LIF estimates and comparing them with SIC products remains largely unchanged, with only minor differences in the results and how they are presented. A few additional caveats have been introduced.

Overall, I find the study to be scientifically valuable, particularly regarding the emulation methodology and the way crossing angles and profiling differences are analyzed. However, I still have quite a few concerns that I believe should be addressed before publication.

Thanks for these comments. In general, we added significant additional analysis, focusing on the dark/specular lead differences as well as strong/weak differences. This led to the generation of new Figures 5 and 6 as well as an additional interpretation of the data to use only specular leads.

**Figure 3 and Its Interpretation**

This figure, which is central to the study, was difficult to follow. The panel labels were incorrect, and axis labels were missing, making interpretation challenging. Additionally, it was not always clear whether the panels referred to a single image (Fig. 2) or all images. Would it be possible to first introduce these concepts using a single-image example, before presenting the full dataset? I read the description multiple times and still found parts of the explanation unclear. Overall, this section felt a bit rushed.

Indeed - both you and Reviewer 1 expressed the same concerns with this section. We have rewritten the entire section, changing notation, and simplifying Figure 3 to illustrate only the points relevant to the discussion. We do not copy the section here, but please see the updates to all in the revised manuscript.

**Validation of LIF at Low Concentrations**

If LIF is to be presented as a product rather than a proof-of-concept, there needs to be a better validation at lower SIC levels. In Part 1 (and the original paper), LIF was only validated on high concentration winter scenes, but here it is applied to multiple months, including SIC down to 15%. The ability of ICESat-2 to profile large regions of open water or low-concentration ice reliably remains uncertain, yet this dataset includes these regions. This discrepancy should be addressed, as it is essential if LIF is intended to be a comprehensive gridded dataset. Ideally, this enhanced validation should have been integrated into Part 1.

The IS2 LIF product is not generally evaluated here in areas with low SIC because of the preprocessing steps we employ. This is captured in Table 1 which shows mean PM-SIC values all exceeding 90% in all months. We discuss this fact in the text: The sea ice areas being intercompared here are highly compact - with a mean SIC for NSIDC-CDR of 98% in summer and 99% non-summer months, reflecting a similar sea ice regime as was examined in Buckley et al. (2024) and the possibility of overestimation of SIC in both seasons. All PM-SIC products indicate a higher ice fraction than the LIF in all seasons. Non-summer biases are similar to that found in OIB data as well as in classified optical data, with a median positive difference of 0.5-2.1% for sea ice that recorded by LIF as being 94.3% ice-covered on average, and 98.2% on average for the NSIDC-CDR PM-SIC product.

**and in the conclusion we highlight**

The LIF product has not yet been validated for low-concentration ice or the CDR-defined marginal ice zones, and its utility in those regions remains an open question, although these areas are critical for understanding overall sea ice variability (Bennetts et al., 2022; Squire, 2022; Horvat, 2022).

**Definitional Differences Between SIC and LIF**

In the response to the original submission, the authors stated that:

"The interpretation of lead type from IS-2 is not the goal of this manuscript, but it is certainly a heavily focused-on problem in the IS-2 community."

This seems like a weak argument, given that lead classification is crucial for LIF estimation and the construction of a gridded LIF dataset. The paper acknowledges that the agreement between LIF and passive microwave SIC is much better when dark leads are excluded. To what extent, then, is this discrepancy simply an issue of definition? Perhaps it would be beneficial to focus on high concentration regions and analyze only specular leads to explore this further.

Thank you for persisting here. We took pains to separate the dark leads and specular leads in the revised manuscript, highlighting that dark leads appear to be contaminated open water segments in two months: July and August. We also note the differences between weak and strong beams in classification of specular leads.

In the abstract, we now include text about the surface type classification uncertainties:

We use these to develop a monthly LIF product, covering up to 46% of the Arctic sea ice cover, which has similar-or-better error qualities compared to PM data, subject to uncertainties in surface type classification associated with surface melting and differences between IS2's weak and strong beams.

Details on the new analysis appear as Section 3.3, where we discuss the decision to exclude dark lead segments and focus on a specular LIF product.

The IS2 surface type field includes two radiometrically-derived classification for open water points: "specular" or "dark" leads. Each could potentially be considered open water segments in this work. Leads in ICESat-2 are identified where the ATL07 segment has a high photon rate, a narrow photon distribution, and Lambertian surface characteristics as determined by the ratio of the photon rate to the background photon rate normalized by the sun elevation. Dark leads are identified as the leads with the

lowest photon rate. These "dark leads" can be at least partially contaminated with both open water and cloudy returns Saha et al. (2024), and are responsible for a significant difference between summer and non-summer LIF data due to known issues in classifying surface meltwater in both PM and IS2 products (Kwok et al., 2019; Tilling et al., 2020; Farrell et al., 2020; Herzfeld et al., 2023). In Figure 5, we plot histograms in summer months that include (gold) or exclude (blue) dark lead segments. We show histograms of the difference in the LIF between the two as Supporting Figure S2 as a function of month. We additionally show in Fig. 6(a) the difference between LIFspec and LIF using all dark leads in summer and non-summer months. These classifications play an important role only in July and August, but not in other months. The impact of dark lead segments on the overall LIF distribution can be seen in Fig. 4, where the shape of the LIF histogram including all dark leads in summer months (gold histogram) is peaked at 81%, with no areas of 100% LIF. On average, including dark leads as open water leads to a reduction in LIF by 9.7% in July and August. By contrast, the specular LIF (blue) is significantly closer to 100% and more closely resembles both the nonsummer LIF values (blue) and those derived from PM algorithms (top row), up to the biases seen in non-summer months. Outside of July and August, the net impact of including dark leads in the LIF calculation is very small, contributing a mean difference in LIF of 0.4% (Fig. 6a, blue histogram).

**Additionally, we discuss the removal of dark leads from the eventual LIF product:**

Because of the potential errors associated with dark lead classification, the similarity in SIC histograms with other PM-SIC products that are known to be biased or more uncertainty in summer months, and the minimal impact of dark leads outside of months with surface melting, we provide here as the core LIF product the one that includes only "specular" leads as open water in all months, which we denote LIFspec. In the comparisons that follow, we also generate an LIF product which masks any grid cells where the dark lead fraction greater than 2.5%, which we term LIFND. The coverage of this reduced dataset is plotted as a dashed black line in Fig.4(a). Eliminating areas with high dark lead fraction reduces LIF coverage by 85% in summer, but just 3% outside of the melt season, and in total reduces LIF extent by 18% by significantly limiting summer intercomparisons.

**Finally, we discuss the "low-hanging fruit" that is improved classification by IS2:**

While we have constrained the errors in LIF arising from uncertain temporal and spatial sampling through emulation, there is significant room to improve the LIF product through surface type classification. This comes about in two ways: first by improving the classification of "dark lead" segments in summer, and second by constraining the differences between weak and strong beam reconstructions of the surface. Typical summer dark lead fractions are 9.7%, and whether this represents melt ponding, surface melt, or open water can be further constrained. The variable inclusion of weak or strong beams alters LIF significantly in all months, due to an approximate doubling of specular leads in the strong beams relative to the weak beams. Both weak-only and strong-only products show an overestimation of SIC by PM products, but the degree

and importance of this overestimation should be further understood and rectified by assessing which of the two accurately depicts the sea ice surface.

For other changes, please see the new Section 3.3.

**Azimuthal Crossings and Beam Configuration**

The introduction of more realistic azimuth-dependent crossings is a good improvement. However, in reality, the ICESat-2 beams are spaced 3 km apart, and weak beams profile the same ice as strong beams with a 90 m separation across track but 2.5 km along-track spacing (ICESat-2 Specs). On L74, the text states the beams are 25 km wide, which seems like a typo? Additionally, the analysis does not appear to account for the fact that three beams have the same orientation—should these crossings be considered differently? On L202, the text mentions requiring 4 crossings, but it is unclear whether this refers to 4 out of the 6 beams or something else.

We fixed the 25km statement as follows:

... however, makes approximately 15 orbits each day, with 3 narrow weak-strong beam pairs spaced evenly across a swath of 6.6 km

We have rewritten references to beam crossings and intersections throughout. Because of the highly constrained azimuthal angles (see Fig 1) as a function of latitude, and the spacing between the beams being larger than the footprint of the OIB measurements, we do not consider beam crossings together.

Here a "crossing" refers to the independent sampling of the sea ice surface by one IS2 beam, whereas an "overflight" refers to a general sampling of the surface by the IS2 satellite - this could lead to as many as 6 "crossings" by the 6 weak and strong beams.

**And later,**

Because the azimuthal angles of beam crossings are heavily constrained as a function of latitude (see Fig.1), we consider each beam in an overflight as an independent sampling of the surface, and below in Sec. 4 we consider differences between weak and strong beams.

**Using ICESat-2 for Internal Cross-Validation**

Given that ICESat-2's multiple beams profile the same ice, this could be used to cross-validate the results internally. Specifically, the authors could compare weak beam results to strong beams and assess whether there is a consistent bias between the two. This would help strengthen the validity of the dataset.

Indeed we now add a comparison of weak and strong beams as Section 3.4, with a new figure 6 that explores the differences in surface classification from weak and strong beams. In general, strong beams report more specular leads, which causes a lower LIF - but both report the same fraction of dark leads. We explain:

IS2 has six separate beams, of which the three strong beams have four times the energy of the weak beam, and consequently four times the photon return and approximately four times the along-track resolution (Markus et al., 2017). The difference in beam energy leads to differences in the classifications of lead segments. We compute summary statistics of LIF data evaluated using strong and weak beams alone in Table 1...

**And discuss the reason for the differences (see Fig 6c),**

The difference between strong and weak beams is caused by an increased fraction of specular lead classifications by the strong beam. In Fig. 6(c) we scatter dark (blue) and specular (orange) lead fraction for the strong-only (y axis) or weak-only (blue axis) LIF data. As in (b), these data are presented only for grid areas where there are more than 11 strong and more than 11 weak beam crossings, a total of 157,000 distinct measurements points. For those points, there is a high correlation ( $r^2 = 0.97$ ) between the dark lead fraction in the two datasets, with the best linear fit (red line, slope 1.06) nearly 1-1 (dashed black line). In contrast, there is still a weaker correlation between respective specular lead fractions ( $r^2 = 0.89$ ), and the best linear fit is closer to 2-1 (blue line, slope 2.18). Out of the 157,000 points, 133,000 (85%) have nonzero dark and specular lead fractions in both strong and weak products. Of these, the median strong beam LIF measurement has a specular lead fraction 5.3% higher than its corresponding weak beam LIF, but the median dark lead fraction difference is just 0.06%.

The difference between weak and strong beams introduces an ambiguity that the current analysis cannot necessarily rectify. We discuss this:

In examining differences between IS2's weak and strong beams, we found that the classification of "dark" leads by weak and strong beams was nearly identical as a portion of overall sea ice segments, but that specular leads were nearly twice as common in strong beam samples than weak beam samples. This leads to consisent weak-strong LIF differences of up to 10% in summer months. Since weak and strong beams are sampling approximately the same sea ice, the difference is likely a consequence of differences in the processing of sea ice surface returns between the two products. The weak-only LIF product aligns with estimates of SIC from PM-SIC products, but with a power and resolution 1/4 that of the strong beams, it is possible that openings in the sea ice cover are missing that are captured by the strong beam. Future work aimed at understanding weak-strong differences in collocated imagery will be important in understanding whether weak beam returns should be disregarded, strong beam retrievals overestimate the fraction of open water along-track, or a combination of both.

**Use of Passive Microwave SIC at High Concentrations**

Perhaps passive microwave SIC should not be used at the highest SIC levels? The analysis could have provided more insight into at which SIC ranges the discrepancies appear and why. As mentioned earlier, using ICESat-2 in low concentration regimes may not be the best approach and I would probably still trust passive microwave more. I do agree the high concentration results are compelling.

We agree with these statements, up to challenges in surface type classification. We explain in the abstract:

We use these to develop a monthly LIF product, covering up to 46% of the Arctic sea ice cover, which has similar-or-better error qualities compared to PM data, subject to uncertainties in surface type classification associated with surface melting and differences between IS2's weak and strong beams.

**and in the text:**

The evaluation of LIF in representing local SIC primarily focused on areas of compact sea ice in Buckley et al. (2024), and because of the preprocessing steps employed in generating the monthly LIF product, nearly all locations of intercomparison in Sec. 3 were also compact ice. For example, as indicated in Table 1, mean NSIDC-CDR in the intercompared regions for producing Figure 5 exceeds 98% - just 0.07% of points had an NSIDC-CDR less than 80%. This limits the degree to which the 15% NSIDC-CDR mask used to define the ATL07 can influence LIF data. The LIF product has not yet been validated for low-concentration ice or the CDR-defined marginal ice zones, and its utility in those regions remains an open question, although these areas are critical for understanding overall sea ice variability (Bennetts et al., 2022; Squire, 2022; Horvat, 2022). The LIF product therefore may provide an independent and possibly improved estimate of SIC in high-concentration, non-melt-affected months, though it has not been examined in areas where the sea ice is low-concentration or highly-variable.

**Code/Data Comments**

Providing the MATLAB Code is great, but how usable and well-documented is it? If this is meant to be a software package, ideally at least one reviewer should test it. I could not find the LIF data at the provided Zenodo link—is this available?

We do hope that this code is easy to follow as we have created it as a plug-and-play setup. We have well-documented code bases for generating gridded IS2 data (which includes the LIF, but also wave and FSD information), running the emulation scheme, and compiling the figures for this manuscript.

We invite the reviewers to run the code, which is meant to be plug-and-play. If requested we can provide a reduced subset of the ATL07 data (which is available through NSIDC) for the reviewers to run. We mirror the entire ATL07 dataset on our local storage, but it is several terabytes of data. Specific Comments

• L21 – Is that statement fully accurate?

We rewrote "1%" to small,

Local errors in PM-SIC are observed to have a compensating effect when integrated over the Arctic or Antarctic, hence the impact of algorithmic uncertainty or bias on estimates of total (Arctic or Antarctic) sea ice area are estimated to be small, even in summer (Notz, 2015; Meier and Stewart, 2019; Kern et al., 2020).

L22 – Reanalyses also provide an alternative method, so perhaps clarify this point.
We added "remote sensing",

Still, no remote sensing alternatives to PM exist for measuring SIC from local to global scales that do not require information about the PM signature of sea ice.

• L31 – Important to emphasize that previous studies (e.g., Kern et al., 2020) already identified these biases.

**We modified the preceding sentence:**

We found that PM-SIC products demonstrated consistent positive biases (1-6%) over compact sea ice, potentially because of the presence of small crack features in the sea ice mosaic that cover a limited portion of the overall surface and are challenging to capture with large PM grid sizes (6.25 to 25 km cells), similar to findings in related studied of PM-SIC and optical data (Kern et al., 2019).

• L34 – Magruder's study states the footprint as 11 m.

**Fixed!**

• L65 – Should also mention dark leads.

**We modified to:**

The construction of LIF relies upon the IS2 ATL07 classification of along-track segments of the ice-ocean surface as being ice or two types of open water: "specular" leads, and "dark" leads. Uncertainty and errors in this classification, which is higher in summer (Tilling et al., 2020; Farrell et al., 2020; Koo et al., 2023), could lead to systematic error in LIF calculations.

• L74 – Possible typo? ICESat-2 has a 6.6 km along-track swath in total.

**Indeed it was, fixed!**

however, makes approximately 15 orbits each day, with 3 narrow weak-strong beam pairs spaced evenly across a swath of 6.6 km

• L82 – Now that the papers are split, Part 1 should better address SIC biases, as there is more space. Could the 2.4% bias be improved using different classification subsets or better aligning IS-2 with imagery?

We view this bias as being a product of imperfect surface classification, since when "perfectly" classified the offset is 1.0%.

• Figure 2 – Could you include the latitude of this example, along with the azimuths being applied?

**We now add:**

The particular image shown in the Figure was acquired on April 7, 2018 north of the Beaufort Sea at 75.51°N, 159.3°W and has a sea ice concentration of 92%. We first take a sample from the appropriate RGT azimuth distribution for this latitude, which at this latitude are approximately at  $8.75^{\circ}$  and  $-9.1^{\circ}$  from due North.

• L114 – Minor point: the term "reference" in "synthetic reference ground track" seems unnecessary, as it is just a line.

This sentence is now modified,

For each angle, we then randomly select a corresponding "tie point" in the image (red dots, a), and draw a straight-line crossing through that tie point at the specified orientation angle (black lines).

• L136 – "The progression from LIF0 to Bi is path-dependent"—this phrasing is quite difficult to understand.

We have now re-written this entire section, for example this sentence now reads:

Because each replicate is different, the progression from the set of single-crossing LIFs,  $LIF_{i,1,k}$ , to  $LIF_{i,1,M} \approx LIF_i^*$  is as well. This means that even when the best-case bias  $B_i^* \sim 0$ , there is uncertainty at smaller values of n associated with the variable convergence to the best-case error.

• L151 – What exactly does P represent here?

We now explain:

With M=100, we select a total number of  $\mathcal P$  unique SGT lists, with  $\mathcal P=1000$  for each image

• Figure 3 – The panel labels are incorrect, making this figure confusing. Please revise.

This is now fixed!

• L177 – Should just be called Arctic, if that's all that is shown.

We now rewrite:

Leveraging the uncertainty information obtained through emulation, we next seek to build an SIC product built from the IS2 LIF. As the data evaluation of Buckley et al. (2024) focused on Arctic scenes, we will focus on Arctic data only - though we do provide Antarctic LIF data in Horvat (2024).

• L202 – In Section 2, you mention requiring 8 total beam intersections, but here it states 4—please clarify.

We now rewrite:

This reduces the number of grid cells over which we develop an LIF product. We combine this restriction with the requirement that the grid cell was intersected by at least 11 separate IS2 beam crossings.

• L218 – Would be good to explicitly reference Part 1 here, as well as previous studies (Kern et al.), which found similar PM SIC biases.

This sentence does not appear in the revised mansucript.

• L220 – The impact of dark lead removal seems more significant in summer than in winter. The phrasing here should be adjusted.

Indeed, please see updated Section 3.

 L221 – Again, this highlights why Part 1 needs a more detailed analysis of dark leads across different seasons.

Agreed! Please see updated Section 3.

**References cited**

- Bennetts, L. G., Bitz, C. M., Feltham, D. L., Kohout, A. L., and Meylan, M. H.: Theory, modelling and observations of marginal ice zone dynamics: Multidisciplinary perspectives and outlooks, Philosophical Transactions of the Royal Society A: Mathematical, Physical and Engineering Sciences, 380, 20210 265, https://doi.org/10.1098/rsta.2021.0265, publisher: Royal Society, 2022.
- Buckley, E., Horvat, C., Yoosiri, P., and Wilhelmus, M. M.: Sea Ice Concentration Estimates from ICESat-2 Linear Ice Fraction. Part 1: Multi-sensor Comparison of Sea Ice Concentration Products with ICESat-2 Data, The Cryosphere Discuss., 2024.
- Cavalieri, D. J., Gloersen, P., and Campbell, W. J.: Determination of sea ice parameters with the NIMBUS 7 SMMR, Journal of Geophysical Research: Atmospheres, 89, 5355–5369, https://doi.org/10.1029/JD089iD04p05355, \_eprint: https://onlinelibrary.wiley.com/doi/pdf/10.1029/JD089iD04p05355, 1984.
- Comiso, J. C. and Sullivan, C. W.: Satellite microwave and in situ observations of the Weddell Sea ice cover and its marginal ice zone, Journal of Geophysical Research, 91, 9663, https://doi.org/10.1029/JC091iC08p09663, 1986.
- Farrell, S. L., Duncan, K., Buckley, E. M., Richter-Menge, J., and Li, R.: Mapping Sea Ice Surface Topography in High Fidelity With ICESat-2, Geophysical Research Letters, 47, https://doi.org/10.1029/2020GL090708, 2020.
- Herzfeld, U. C., Trantow, T. M., Han, H., Buckley, E., Farrell, S. L., and Lawson, M.: Automated Detection and Depth Determination of Melt Ponds on Sea Ice in ICESat-2 ATLAS Data—The Density-Dimension Algorithm for Bifurcating Sea-Ice Reflectors (DDA-Bifurcate-Seaice), IEEE Transactions on Geoscience and Remote Sensing, 61, 1–22, https://doi.org/10.1109/TGRS.2023. 3268073, conference Name: IEEE Transactions on Geoscience and Remote Sensing, 2023.
- Horvat, C.: Floes, the marginal ice zone and coupled wave-sea-ice feedbacks, Phil.Trans.R.Soc.A, 380, https://doi.org/10.1098/rsta.2021.0252, 2022.
- Horvat, C.: ICESat-2 Emulator Code, Github, https://doi.org/10.5281/zenodo.13549563, 2024.
- Kern, S., Lavergne, T., Notz, D., Pedersen, L. T., Tonboe, R. T., Saldo, R., and Soerensen, A. M.: Satellite Passive Microwave Sea-Ice Concentration Data Set Intercomparison: Closed Ice and Ship-Based Observations, The Cryosphere, pp. 1–55, https://doi.org/10.5194/tc-2019-120, 2019.

- Kern, S., Lavergne, T., Notz, D., Pedersen, L. T., and Tonboe, R.: Satellite passive microwave sea-ice concentration data set inter-comparison for Arctic summer conditions, The Cryosphere, 14, 2469–2493, https://doi.org/10.5194/tc-14-2469-2020, 2020.
- Koo, Y., Xie, H., Kurtz, N. T., Ackley, S. F., and Wang, W.: Sea ice surface type classification of ICESat-2 ATL07 data by using data-driven machine learning model: Ross Sea, Antarctic as an example, Remote Sensing of Environment, 296, 113726, https://doi.org/10.1016/j.rse.2023. 113726, 2023.
- Kwok, R., Cunningham, G., Markus, T., Hancock, D., Morison, J., Palm, S. P., Farrell, S. L., Ivanoff, A., Wimert, J., and Team, t. I.-. S.: ATLAS/ICESat-2 L3A Sea Ice Height, Version 1. Boulder, Colorado USA., Tech. rep., NSIDC, Boulder, Colorado USA, https://doi.org/https://doi.org/10.5067/ATLAS/ATL07.001, 2019.
- Kwok, R., Petty, A., Cunningham, G., Markus, T., Hancock, D., Ivanoff, A., Wimert, J., Bagnardi, M., Kurtz, N., and Team, t. I.-. S.: ATLAS/ICESat-2 L3A Sea Ice Height, Version 6., Tech. rep., NASA National Snow and Ice Data Center Distributed Active Archive Center, Boulder, Colorado USA, https://doi.org/https://doi.org/10.5067/ATLAS/ATL07.006, issue: May, 2023.
- Lavergne, T., Sørensen, A. M., Kern, S., Tonboe, R., Notz, D., Aaboe, S., Bell, L., Dybkjær, G., Eastwood, S., Gabarro, C., Heygster, G., Killie, M. A., Brandt Kreiner, M., Lavelle, J., Saldo, R., Sandven, S., and Pedersen, L. T.: Version 2 of the EUMETSAT OSI SAF and ESA CCI sea-ice concentration climate data records, The Cryosphere, 13, 49–78, https://doi.org/10.5194/tc-13-49-2019, 2019.
- Markus, T., Comiso, J. C., and Meier, W.: AMSR-E/AMSR2 Unified L3 Daily 25.5 km Polar Gridded Brightness Temperatures, Sea Ice Concentration, & Depth, Version 1, https://doi.org/10.5067/TRUIAL3WPAUP, 2017.
- Meier, W.: AMSR-E/AMSR2 Unified L3 Daily 25.5 km Polar Gridded Brightness Temperatures, Sea Ice Concentration, & Depth, Version 1, https://doi.org/10.5067/TRUIAL3WPAUP, tex.doi+duplicate-1: 10.5067/TRUIAL3WPAUP, 2018.
- Meier, W. N. and Stewart, J. S.: Assessing uncertainties in sea ice extent climate indicators, Environmental Research Letters, 14, 035 005, https://doi.org/10.1088/1748-9326/aaf52c, 2019.
- Meier, W. N., Peng, G., Scott, D. J., and Savoie, M. H.: Verification of a new NOAA/NSIDC passive microwave sea-ice concentration climate record, Polar Research, 33, 21 004, publisher: Taylor & Francis, 2014.
- Meier, W. N., Fetterer, F., Windnagel., A. K., and Stewart, J. S.: NOAA/NSIDC Climate Data285 Record of Passive Microwave Sea Ice Concentration, Version 4, Tech. rep., NSIDC: National Snow and Ice Data Center, Boulder, Colorado USA, https://doi.org/10.7265/efmz-2t65, 2021.
- Notz, D.: How well must climate models agree with observations?, Philosophical Transactions of the Royal Society A: Mathematical, Physical and Engineering Sciences, 373, 20140 164, https://doi.org/10.1098/rsta.2014.0164, 2015.

- Saha, M., Kurtz, N. T., Wimert, J., and Palm, S.: Improving near-coastal lead classification and freeboard measurements from ICESat-2, vol. 2024, pp. C13C-0557, URL https://ui.adsabs.harvard.edu/abs/2024AGUFMC13C.0557S, aDS Bibcode: 2024AGUFMC13C.0557S, 2024.
- Spreen, G., Kaleschke, L., and Heygster, G.: Sea ice remote sensing using AMSR-E 89-GHz channels, Journal of Geophysical Research, 113, C02S03, https://doi.org/10.1029/2005JC003384, 2008.
- Squire, V. A.: Marginal ice zone dynamics, Philosophical Transactions of the Royal Society A: Mathematical, Physical and Engineering Sciences, 380, 20210 266, https://doi.org/10.1098/rsta. 2021.0266, publisher: Royal Society, 2022.
- Tilling, R., Kurtz, N. T., Bagnardi, M., Petty, A. A., and Kwok, R.: Detection of Melt Ponds on Arctic Summer Sea Ice From ICESat-2, Geophysical Research Letters, 47, 1–10, https://doi.org/10.1029/2020GL090644, 2020.

---

## Author Response (AR2)

Dear Dr. Howell,

Thanks for your and the reveiwer efforts. We have addressed the minor comments from reviewers - importantly, the horrendous-looking Figure 6 (which escaped us during review of the submission) was re-plotted and looks as intended.

Best,

Christopher Horvat on behalf of the authors.

**Reviewer 1**

• Figure 4 is much better now, and I think it is acceptable to use this to explain the peak in dark leads during July and August. However, this could be more clear if you show the individual monthly points instead of the dashed line. I think this is an important part of the analysis and the figure doesn't make it clear. A mean seasonal cycle would maybe be easier to justify this too.

This seems to refer to the LIF $_{ND}$  plot which is as a dashed line. We now replot Figure 4 with the monthly data with scattered dots to show the differing values there, since these are only two months. The revised caption reads:

- (a) Arctic sea ice extent of 6 PM-SIC products (dashed colored lines) compared to the area well-sampled by IS2 (black line, black scatter) from October 2018-December 2023. Black line with blue scatters is the IS2 extent when excluding areas with more than 2.5% dark lead fraction, LIF $_{ND}$ . "Summer months" have red background.
- Figure 6 obviously needs a lot of work to improve readability, that was a little disappointing.

Yes, this was an embarassing oversight that must have happened on generating a final figure file - the version that appeared in the submission is not what we have locally or should be produced. Please see the much-more-legible Figure 6 in the revised submission.

**Reviewer 2**

- This paper describes a thorough analysis of errors associated with a new linear ice fraction product from ICESat-2 in comparison with sea ice concentration products from several passive microwave products. The linear ice fraction product has potential to enhance current sea ice concentration data products.
  - The revised manuscript is much improved. The authors have addressed the concerns raised in my initial review to my satisfaction. I have a couple of minor technical comments for clarification noted below. Otherwise, I recommend this paper for publication in The Cryosphere.
- P11, L243: The two AMSR2 sea ice concentration products are stated here, but why are they presented as either/or in this sentence? Maybe this needs clarification or just a revision to the sentence.

Thanks - we rewrote this sentence as we use both products.

We also include two algorithms using brightness temperature data from the Advanced Microwave Scanning Radiometer 2 (AMSR2) sensor on board the JAXA GCOM-W satellite, computed using (5) the NASAteam2 algorithm (Meier, 2018), and (6) the ASI-ARTIST algorithm Spreen et al. (2008).

• P11, L254: The NASA sea ice concentration algorithm for AMSR2 is the NASA Team 2 algorithm. I would abbreviate this product as AMSR2-NT2 throughout to paper to clarify that it is not the NASA Team (NT) algorithm applied to AMSR2 brightness temperatures. Related to this abbreviation, I noticed the labels for AMSR2-NT[2] and AMSR2-ASI in Figure 4 have typos.

This is a good point. We now fix throughout, and in the figure change AMRS2 to AMSR2.

**References cited**

Meier, W.: AMSR-E/AMSR2 Unified L3 Daily 25.5 km Polar Gridded Brightness Temperatures, Sea Ice Concentration, & Depth, Version 1, https://doi.org/10.5067/TRUIAL3WPAUP, tex.doi+duplicate-1: 10.5067/TRUIAL3WPAUP, 2018.

Spreen, G., Kaleschke, L., and Heygster, G.: Sea ice remote sensing using AMSR-E 89-GHz channels, Journal of Geophysical Research, 113, C02S03, https://doi.org/10.1029/2005JC003384, 2008.